# Stabilization of Calcium Oxalate Precursors during the Pre- and Post-Nucleation Stages with Poly(acrylic acid)

**DOI:** 10.3390/nano11010235

**Published:** 2021-01-18

**Authors:** Felipe Díaz-Soler, Carlos Rodriguez-Navarro, Encarnación Ruiz-Agudo, Andrónico Neira-Carrillo

**Affiliations:** 1Programa de Doctorado en Ciencias Silvoagropecuarias y Veterinarias, Campus Sur Universidad de Chile, Santa Rosa 11315, La Pintana, Santiago 8820808, Chile; fdiazsoler@gmail.com; 2Department of Biological and Animal Science, University of Chile, Santa Rosa 11735, La Pintana, Santiago 8820808, Chile; 3Department of Mineralogy and Petrology, University of Granada, Fuente nueva S/N, 18002 Granada, Spain; carlosrn@ugr.es (C.R.-N.); encaruiz@ugr.es (E.R.-A.)

**Keywords:** calcium oxalate, pre-nucleation clusters, poly(acrylic acid), amorphous calcium oxalate (ACO)

## Abstract

In this work, calcium oxalate (CaOx) precursors were stabilized by poly(acrylic acid) (PAA) as an additive under in vitro crystallization assays involving the formation of pre-nucleation clusters of CaOx via a non-classical crystallization (NCC) pathway. The in vitro crystallization of CaOx was carried out in the presence of 10, 50 and 100 mg/L PAA by using automatic calcium potentiometric titration experiments at a constant pH of 6.7 at 20 °C. The results confirmed the successful stabilization of amorphous calcium oxalate II and III (ACOII and ACO III) nanoparticles formed after PNC in the presence of PAA and suggest the participation and stabilization of polymer-induced liquid-precursor (PILP) in the presence of PAA. We demonstrated that PAA stabilizes CaOx precursors with size in the range of 20–400 nm. PAA additive plays a key role in the in vitro crystallization of CaOx stabilizing multi-ion complexes in the pre-nucleation stage, thereby delaying the nucleation of ACO nanoparticles. Indeed, PAA additive favors the formation of more hydrated and soluble phase of ACO nanoparticles that are bound by electrostatic interactions to carboxylic acid groups of PAA during the post-nucleation stage. These findings may help to a better understanding of the pathological mineralization resulting in urolithiasis in mammals.

## 1. Introduction

Calcium oxalate (CaOx) represents the main inorganic component of kidney stones, a pathology known as urolithiasis [1,2]. Urolithiasis affects 7–13% of North America population, and between 5–9% and 1–5% of the population in Europe and Asia, respectively [3]. Indeed, it has a 50% recurrence rate [4]. The formation of kidney stone is the result of a combination of physicochemical factors such as urinary fluid supersaturation and an imbalance of macromolecules that act as promoters or inhibitors of in vivo crystallization of CaOx [5,6,7]. As it is the case for both biogenic and non-biogenic materials, the formation of CaOx involves the stages of nucleation and crystal growth [8]. In nature, CaOx (CaC_2_O_4_·nH_2_O, where n = 1 to 3) has three hydrated crystalline forms: monohydrate (COM, or whewellite), dihydrate (COD or weddellite) or trihydrate (COT or caoxite). However, only COM and COD are found in urine, whereas COM, the most thermodynamically stable form, is the main inorganic crystal found in kidney stone [9,10,11,12].

Recent experimental evidence suggests that the nucleation of CaOx occurs via a non-classical crystallization (NCC) pathway, which involves the bonding of calcium (Ca^2+^) and oxalate (C_2_O_4_^2−^) ions into multi-ion associates known as pre-nucleation clusters (PNC), which can evolve into a condensed phase, amorphous CaOx (ACO), which later on gives rise to the different types of crystalline forms of CaOx [13,14,15]. The foregoing path involves a multi-step crystallization process, unlike classical crystallization (CC), that predicts that at a sufficiently high supersaturation, nucleation of a stable crystalline seed will occur which will continue to grow via incorporation of growth units (ions or molecules). It has been pointed out that PNC can be stabilized in the presence of citrate molecule used as an additive during in vitro CaOx precipitation using computer-controlled automatic calcium potentiometric titration [14]. Using organic additives, it is also experimentally possible to stabilize ACO nanoparticles and prevent the kinetic crystallization of CaOx. Anionic molecules such as citric acid can alter the crystallization pathway of CaOx by modulating the water content of ACO nanoparticles, thus in the presence of citric acid, three amorphous hydrated forms have been obtained which have been referred to as ACO I, ACO II and ACO III, giving rise to the crystalline hydrates COM, COD and COT, respectively [15].

On the other hand, numerous organic additives with different ionic entities, including natural macromolecules, synthetic polymers and proteins have been studied from a CC point of view [16,17,18,19]. The main focus of these studies has been the characterization of the role played by urinary macromolecules, because these are the main organic component of the kidney stone matrix [20]. Extensive evidence demonstrates that acidic macromolecules have an inhibitory effect on the nucleation and growth of CaOx [21,22,23,24]. Natural anionic macromolecules as well as synthetic analogs such as poly(acrylic acid) (PAA) have been shown to inhibit the nucleation, aggregation and growth stages of COM and the results have been interpreted according to CC theory [24,25]. However, it has been suggested that PAA would allow the formation of ACO by a PILP route [25], in a similar manner to that described for calcium carbonate [26,27] or calcium phosphate [28], but this has not yet been proved. The participation of dense liquid precursor phases during crystallization represents an example of a non-classical multistage crystallization pathway, that involves the formation of a dense liquid following a liquid-liquid phase separation in supersaturated solutions. These are highly hydrated precursor phases with fluidic character that adopt an emulsion-like conformation in solution. Nowadays, the effect of acidic macromolecules on the formation of precursor phases and in the development of nanostructures during the early stages of CaOx crystallization is largely unknown.

In this study, we show how acidic macromolecules influence the dynamics and kinetics of CaOx crystallization by exerting control on the pre- and post-nucleation stages by delaying the crystallization of solid phases and allowing the stabilization of PNCs. The aim of this paper was to determine the effect of PAA concentration (10, 50 and 100 mg/L) on ACO structures during precipitation, using a computer-controlled automatic titration system, at a constant pH of 6.7 at 20 °C. PNCs as well as liquid and ACO precursor structures were identified in the presence of PAA. The characterization of the phases formed in the pre- and post-nucleation stages was carried out using dynamic light scattering (DLS), transmission electron microscopy (TEM) with selected area electron diffraction (SAED), Raman and FTIR spectroscopy, field emission scanning electron microscopy (FESEM) and X-ray diffraction (XRD) techniques.

## 2. Materials and Methods

### 2.1. Reactants and Instruments

Na_2_C_2_O_4_ (99%), CaCl_2_·2H_2_O (99%), poly(acrylic acid) (PAA, powder with Mw 5100 Da), NaOH (≥98%) and H_2_C_2_O_4_ (≥99%) were acquired from Sigma-Aldrich (St. Louis, MO, USA). All experiments and solution preparations were performed using fresh Milli-Q water (LabostarTM TWF, Evoqua Water Technologies LLC, Warrendale, PA, USA). Absolute ethanol used here was of analytical grade. Crystallization experiments of CaOx were carried out using the Titrando 907 instrument, Metrohm No. 2,905.0020, (Metrohm AG, Herisau, Switzerland) using the commercial computer software Tiamo™, version 2.3. During the entire pre- and post-nucleation experiment, continuous free Ca^2+^ monitoring was performed using a calcium ion selective electrode (ISE-Ca^2+^), and pH measurements were performed with a glass electrode (Metrohm). Moreover, the continuous conductivity monitoring of samples was carried out using a 5-ring conductivity measurement cell (Metrohm). Particle size distribution (PSD) analysis was performed by dynamic light scattering (DLS) with a Zetasizer Nano zs instrument equipped with a 633 nm He-Ne Laser, 4 mW (Malvern instruments Ltd., Malvern, Worcestershire, UK). Volume based PSD of nanoparticles was determined by using a refractive index of 1.330, and a viscosity of 0.8872 cP in aqueous media at 25 °C. Zeta potential measurements of nanoparticles formed during all nucleation stages were performed using a Stabino Zeta Check instrument. The CaOx precursors and nanoparticles obtained were analyzed by transmission electron microscopy (TEM) in a Hitachi HT7700 (Hitachi, Tokyo, Japan) operated at 80 kV and in a FEI Titan (FEI company, Hillsboro, OR, USA) operated at 300 kV. Selected area electron diffraction (SAED) was used to determine the crystalline nature of the precipitates. The structural analysis and chemical composition of CaOx precipitates obtained in the post-nucleation stage was performed by field emission scanning electron microscopy (Zeiss, Jena, Germany) with energy dispersive X-ray spectrometry (FESEM-EDS, Zeiss SUPRA40VP), X-ray diffraction (XRD, Philips X’Pert Pro X-ray; CuKα radiation, with a range of 2θ of 2–80°, Malvern Panalytical, Malvern, UK), Fourier transform infrared spectroscopy (Jasco FT/IR 6600, Jasco corporation, Tokyo, Japan), and Raman spectroscopy (Advantage Systems, DeltaNu, Laramie, Wyoming, USA) equipped with a 785 nm laser.

### 2.2. CaOx Crystallization

A solution of 20 mM CaCl_2_ was added at a rate of 60 µL/min to 100 mL of a 5 mM Na_2_C_2_O_4_ solution prepared in the absence and presence of PAA (10, 50 and 100 mg/L) at pH 6.7. Throughout the experiment, free Ca^2+^ monitoring was performed using an ISE-Ca^2+^ and pH measurements at a constant temperature of 20 °C. To maintain a constant pH of 6.7 during the entire period of the experiment, the equipment added a 0.1 M NaOH solution at a variable rate (automatic burette) according to the test requirements. To avoid the matrix effect or minimize it when using ISE-Ca^2+^, a calibration curve was performed for each crystallization assay. The calibration was performed by adding (using an automatic burette) 20 mM CaCl_2_ at a rate of 60 µL/min to a 100 mL NaCl solution, which had the same ionic strength as the Na_2_C_2_O_4_ solution used in each experimental essay. Thus, the experimental values obtained using the ISE-Ca^2+^ electrode were adjusted to the calibration curve carried out prior to each titration test in the absence and presence of PAA. The size of nanoparticles obtained 250 s prior to the nucleation stage for each trial was determined using DLS (Malvern instruments Ltd., Malvern, Worcestershire, UK). All titration experiments were performed in triplicate.

### 2.3. Conductivity Measurements of the Ionic Complexes of Ca:C_2_O_4_

Continuous conductivity measurements were carried out for calcium and oxalate ions systems using a 5-ring conductivity measurement cell (Metrohm) and the ionic complexes Ca:C_2_O_4_ ratio was determined through Kohlrausch’s law. This law uses independent ion migration, by determining the molar conductivity of the ionic complexes formed and the difference between the measured conductivity and the theoretical conductivity considering the control titration tests assuming an ideal solution.

### 2.4. Characterization of Pre-Nucleation Entities

Samples from solution were obtained at 250 s prior to the nucleation stage. Right after sample collection was carried out, quenching in ethanol was performed followed by centrifugation (7600 g), and washing (twice) with ethanol. Then, the collected samples were stored in acetone. TEM-SAED was used to analyze the resultant nanoparticles presumably formed after dehydration of PNC aggregates or a dense liquid precursor phase.

### 2.5. Characterization of Post-Nucleation Stage Precipitates

Each crystallization sample was obtained at 500, 1200 and 6000 s after the onset of the nucleation stage. This was performed to determine the influence of PAA on the amorphous and the crystalline phases of CaOx. Right after sample collection was carried out, quenching in ethanol and drying were performed as before prior to TEM-SAED analysis. After that, the samples were observed by FESEM and their chemical composition analyzed by EDS. In addition, for determining the molecular nature, as well as crystalline and amorphous features of the resultant particles, the samples were analyzed by FTIR-ATR, Raman spectroscopy and XRD. In addition, freezing of the solution samples with liquid nitrogen and subsequent lyophilization in the absence and presence of PAA was performed prior to XRD analysis.

## 3. Results and Discussion

### 3.1. Non-Classical Crystallization of CaOx

The crystallization kinetics of CaOx and the evolution of free Ca^2+^ concentration in the absence and presence of PAA in titration experiments of CaOx and pure water is shown in Figure 1. In general, the pre-nucleation stage is characterized by a sustained increase of the free Ca^2+^ concentration. Then the nucleation stage is visualized by the fall in the free Ca^2+^ concentration, to subsequently produce a stabilization of the free Ca^2+^ concentration detected in post-nucleation stages, which correlates with the solubility of the solid phases formed. In the pre-nucleation stage, an evident difference between added Ca^2+^ and detected free Ca^2+^ concentrations both in the absence and presence of PAA was observed, even in the very early stages of the nucleation essay. The possibility that the formation of ionic pairs generates such a marked reduction in the concentration of the free Ca^2+^ especially in the initial stages of the test and in unsaturated solutions is unlikely, as has been pointed out during the formation of ionic pairs in calcium carbonate [29], which indicates that the decrease on calcium ionic activity is due to the association of Ca^2+^ and C_2_O_4_^2−^ entities (i.e., formation of PNCs). Moreover, PNCs present a rate of constant formation reflected by the linear increase of free Ca^2+^ concentration detected in the kinetic crystallization curves. Ionic CaOx complexes have been described in previous studies [14,15]. Using the methodology outlined by Ruiz Agudo et al., we determined the concentration of C_2_O_4_^2−^ ions bonded in PNCs, as well as those free in solution using conductivity measurements. Remarkably, the concentration of bonded C_2_O_4_^2−^ determined using conductivity and bonded Ca^2+^ determined from the ion selective electrode measurements was in perfect agreement when considering a C_2_O_4_:Ca binding ratio of 2:1 for the PNC (Appendix A). This result reflects that ionic species other than ionic pairs are formed in our system, which is in agreement with previous studies [14]. In addition, we observe that the presence of PAA delays the nucleation of CaOx. This effect is more marked by increasing the concentration of PAA, which can shift the nucleation point from 680 s in the absence of this additive to values of 980, 1360 and 1850 s using PAA concentrations of 10, 50 and 100 mg/L, respectively (Figure 1a). These findings suggest that PAA stabilizes PNCs, preventing their aggregation, thereby exerting its inhibitory effect on CaOx nucleation. In addition, it is observed that the supersaturation of the system in the presence of PAA is greater than that observed in the control tests at the beginning of nucleation. This can be explained by the stabilization of the PNC, which allows both a higher concentration of PNC prior to nucleation and a greater concentration of free Ca^2+^ ions in the system. Furthermore, interaction between carboxylic groups of PAA with free Ca^2+^ ions, can lead to a significant concentration decrease, an effect that was evaluated by monitoring the concentration of free Ca^2+^ in the presence of PAA dissolved in pure water (Figure 1b). As observed, the decrease in concentration of the free Ca^2+^ ions in the oxalate-free solutions was evidently lower than that experimentally determined by titration essays in the presence of oxalate. The difference observed in these two experiments can be explained by the incorporation of Ca^2+^ ions into the PNCs, leading to a decrease in the concentration of free Ca^2+^ detected. These evidences support that the nucleation delay produced by PAA, is through the stabilization of PNCs rather than by a decrease in the supersaturation in the solution produced by the complexation of Ca^2+^ ions by carboxyl groups of PAA. This effect is clearly observed when comparing the titration curves obtained for the 10 mg/L PAA runs in water and in the presence of C_2_O_4_^2−^. Figure 1b shows that in water the complexation effect of 10 mg/L is very limited, as shown by the small difference between dosed and free Ca^2+^. In contrast, in the presence of oxalate ions, a marked reduction in the free Ca^2+^ is observed than cannot be explained by a complexation effect. Moreover, the delay in the onset of nucleation in the presence of 10 mg/L is associated with the fact that nucleation takes place at a free Ca^2+^ value double than that of the control.

This behavior is also observed in the case of runs performed using 50 mg/L and 100 mg/L PAA, where Ca-complexation increases with PAA content. At the highest PAA concentration tested here (100 mg/L), the CaOx kinetic crystallization curve acquires a concave shape during the first 600 s, which indicates a complexation of Ca^2+^ ions by PAA molecules, which markedly decreases the free Ca^2+^ concentration during the early stages of the titration experiment. However, a flattening of the slope of the linear segment of the kinetic crystallization curve in the pre-nucleation stage is also observed, which is not related to complexation of Ca^2+^ by PAA, but to an enhanced incorporation of Ca^2+^ into the PNCs.

Afterwards, the presence of PAA induces the formation of a more soluble solid phase, which is reflected by a stabilization of the free Ca^2+^ concentration in the post-nucleation stage. The latter is corroborated by determining the ion activity product (IAP) for each crystallization test (Figure 2). The IAP results were obtained by direct measurements of free calcium obtained by ISE and the indirect determination of the concentration of oxalate ions for each essay using the conductivity values and the Kohlrausch’s law. Herein, we observed an increase in the value of IAP when PAA was used, reflecting a concentration dependent effect on the solubility of the precipitated solids.

### 3.2. Characterization of Pre-Nucleation Products

The characterization of the entities formed during the pre-nucleation stage in the presence and absence of PAA was performed by TEM-SAED analysis (Figure 3). Note that the observed solids are the result of the drying procedure to which the solution aliquots were subjected: they therefore correspond to solid structures, now ACO nanoparticles, probably formed after drying of PNC aggregates (or dense liquid precursor phases). Figure 3a,b shows the TEM images of ACO obtained in the absence of PAA, where a 200 nm structure is observed, which is formed by a narrow grouping of 10 to 20 nm nanoparticles that have different electron-density, reflected in contrast variations. In addition, fused ACO nanoparticles with a clear loss of the shape of their boundaries were observed forming heterogeneous and elongated nanoparticles, suggesting a possible liquid nature of these structures (i.e., they seem to correspond to a dried dense liquid precursor phase). They coexist with solid nanoparticles 10 to 20 nm in size, characterized by a well-defined circular structure. The SAED pattern confirms the amorphous nature of such structures. On the other hand, Figure 3c–f shows the TEM images of nanoparticles obtained in the presence of 10, 50 and 100 mg/L of PAA, respectively. Our results indicate that ACO nanoparticles from 20 to 250 nm in size are ungrouped and only occasionally there is a close interaction between two or more of them (Figure 3c). The resultant nanoparticles showed different electron-density. Moreover, when 50 mg/L PAA was used, pre-nucleation structures were formed by the fusion of multiple nanoparticles, forming a floc-like structure reaching a particle size of 1 µm.

The fusion of nanoparticles with no clear boundary between them, linked through a less dense and poorly structured phase, was observed, and only the presence of a diffuse neck was seen (Figure 3d), giving an emulsion-like appearance. Overall, these features can be associated with a late liquid phase formed after its densification by expulsion of water, suggesting an originally liquid nature of these structures [26,30]. In addition, dense (dark) particles were observed that contained nanostructures having a lower degree of electron-density (Figure 3c,f). Moreover, in the presence of 100 mg/L PAA, heterogeneous forms of ACO were obtained, showing a spheroidal or ellipsoidal shape with particle sizes up to 400 nm with low variation in contrast between the center and their periphery (Appendix A). Nanoparticles with drop-like shapes that present a low variation in contrast between the center and their periphery have also been associated with liquid-like characteristics [30]. Although the characterization of liquid precursor phases is difficult to carry out without the use of cryogenic techniques, it has been shown that there are no significant differences between the structural characteristics of the amorphous particles obtained using Cryo-TEM and conventional TEM [31]. In this way this work identifies solid structures that maintain the characteristics of liquid precursors by using conventional TEM. These experimental results suggest that PAA favors and stabilizes the formation of a liquid-like or PILP precursor phase finally leading to ACO nanoparticles formation during its drying. It should be noted that the smallest entities observed here, a few nanometers in size, are considered as solid ACO probably from dried PNC aggregates resulting from the quenching and drying procedure adopted here. The formation of larger particle aggregates likely resulted from the drying of PILP. Our results indicate that solid ACO nucleation is preceded by the formation of PNCs which aggregate forming possibly a PILP, which is favored by the presence of an acidic macromolecule. Thus, we suggest that nucleation may occur through the densification of liquid precursor entities and their subsequent aggregation, which leads to the formation of amorphous nanoparticles, which in turn may allow crystal growth via a non-classical aggregation mechanism [8] (see below). As it has been suggested for calcium carbonate, the transformation process for CaOx could involve the densification of the liquid phase formed after PNCs by exclusion of water (an entropically-driven process) or the nucleation of ACO within the dense liquid phase [26,27,32].

In addition, PAA allows the stabilization of the precursor phase(s), including ACO nanoparticles, which is reflected by the poor aggregation of the nanoparticles observed in samples collected during the pre-nucleation stage. On the other hand, ACO formed in the presence of PAA at the pre-nucleation stage after drying of collected samples showed Zeta potential values more negative (−36 ± 0.2 mV) than those of ACO obtained without PAA (−18 ± 0.2 mV) (Appendix A). These results show that PAA has interacted with or has been adsorbed on such nanoparticles and suggest that PAA prevents the agglomeration of PNC and ACO nanoparticles possibly by electrostatic repulsion due to its highly negative charge (at the experimental pH) and by the steric hindrance of neighboring non-polar chains in PAA. The ultimate effect of such interactions is the observed delay of the onset of the post-nucleation stage. Moreover, DLS analysis (Figure 4) shows that ACO nanoparticles formed at the very early stages of nucleation in nearly all titration experiments, both in the absence and presence of PAA, displayed a maximum at ~160 nm, consistent with the size of nanoparticle aggregates observed using TEM. However, the control run showed an additional maximum at ~60 nm, which should correspond to the individual ACO nanoparticles (or incipient aggregates of smaller 10–20 nm nanoparticles observed with TEM). Interestingly, when PAA was dosed at 100 mg/L, bigger particles reaching sizes up to 400 nm were observed. Such particles were detected during the very early stages of nucleation when the supersaturation of the solution was high. One possibility is that these nanoparticles are covered extensively by PAA, which could favor the establishments of a bridging flocculation process which can favor liquid-liquid phase separation and, consequently, the formation of larger PILP structures.

### 3.3. XRD Analysis of Precipitates

Samples collected during the post-nucleation stage were analyzed by XRD (Figure 5) to study the influence of PAA on the metastability of amorphous phases of CaOx and on the development of different crystalline phases. The XRD diffractograms of solids formed in the absence of PAA initially showed a rapid formation of a crystalline solid phase which is characterized by a low intensity of Bragg reflections, which after 6000 s showed an intensity increase, presenting characteristic Bragg reflections of hydrated COM (Figure 5a). The presence of PAA directs crystallization kinetics towards COD and COT hydrates that are thermodynamically less stable. However, the concentration of PAA determines the precipitated crystalline phase. A concentration of 10 mg/L PAA, led to the initial formation (500 and 1200 s post-nucleation) of COT. Indeed, broad Bragg reflections of very low intensity were observed, suggesting the initial formation of COT hydrate with very low crystallinity. However, after 6000 s post-nucleation, hydrated forms of mature COD and COT were observed (Figure 5b). These results suggest that PAA at concentration of 10 mg/L stabilizes and favors the precipitation of ACO III in the very early nucleation stage, and slowly evolves to COT hydrated form after 500 and 1200 s post-nucleation. When PAA was dosed at a concentration of 50 mg/L, crystalline precipitates were initially observed, and they presented some low intensity Bragg reflections of COD, which evolve to mature COD and COT after 6000 s post-nucleation (Figure 5c). This suggests that PAA at this concentration favors the precipitation of ACO II in the very early nucleation stage, which after 500 s post-nucleation evolves partially to COD. When PAA was dosed at 100 mg/L, an early development of COD was observed and after 6000 s post-nucleation stage, the coexistence of COD and COT was detected (Figure 5d). Our results clearly demonstrate that PAA induces the early formation of COD and COT, probably favoring the precipitation of metastable amorphous precursor phases of ACO II and III, which both later evolve to the hydrated COD and COT crystalline forms. Note that the latter two crystalline hydrates are thermodynamically less stable than COM and are less active in the formation of kidney stones. In turn, as seen in Figure 2, they present a higher solubility when crystallization of CaOx occurs in the presence of PAA. Thus, a decrease in acidic macromolecules synthesized at the renal tubule level could play a key role in the formation of kidney stones and / or the incorporation of synthetic macromolecules in the urinary tract could have a beneficial role in preventing this pathology.

### 3.4. FTIR Analysis of Precipitates

The comparison of the FTIR spectra of CaOx particles obtained in the absence and presence of PAA is shown in Figure 6. The FTIR spectra show great similarity in the stretching vibrations of C=O at 1600 cm^−1^, indicating that PAA does not modify its geometry upon interaction of Ca^2+^ with C_2_O_4_^2−^ ions. However, the CaOx precipitates obtained in the presence of PAA (Figure 6b–d) show that the stretching vibration of C=O at 1384 cm^−1^ and the stretching vibrations of C–O at 945, 885 and 654 cm^−1^ of the fingerprint zone characteristic of COM are attenuated (Figure 6b) or disappear (Figure 6c,d) proportionally to the concentration of PAA, yielding a spectrum that is compatible with COD. These findings indicate that PAA favors the formation of COD over COM, which correlates well with the results obtained by XRD and Raman spectroscopy (Appendix A). In turn, the presence of PAA produces differences in the absorption band due to the coordinated water. In the absence of PAA, multiple weak absorption bands (O–H stretching) are observed between 3000 and 3500 cm^−1^. However, in the presence of PAA, a broad and intense absorption band between 3000 and 3500 cm^−1^ is observed with two individual absorption bands at 3144 and 3500 cm^−1^. Therefore, the presence of this intense broad band suggests that there are more coordinated water molecules in the COD and COT phases (consistent with their stoichiometry) but having a lower degree of ordering.

### 3.5. Ultrastructural Characterization of Precipitates

The ultrastructural characterization of the resultant precipitates during the post-nucleation stage was carried out by TEM-SAED in conjunction with FESEM-EDS (Figure 7 and Figure 8). The precipitates obtained in the absence of PAA are made up of highly aggregated nanostructures, composed of both spheroidal and pseudo-hexagonal nanocrystals 50 to 400 nm in size (Figure 7a,b), which are classically associated with COM. These structures exhibited different degrees of crystallinity when SAED analysis was performed and are in accordance with XRD analysis. The SEM images show large particles of 2 to 3 µm obtained in the early stages of nucleation, with a clear pseudo-hexagonal crystal shape, in addition to the ACO nanoparticles of 200 to 500 nm. The particles have a nanogranular texture (Figure 7c,d), as observed in numerous biominerals, and some elongated particles (resembling a crystalline phase) are covered by spherical nanoparticles (Figure 7d) [33]. This probably indicates that the mechanism of crystal growth involves the attachment of ACO nanoparticles rather than classical monomer by monomer growth, as also observed in calcite biominerals and its biomimetics [34].

The precipitates obtained in the presence of PAA are disaggregated and exhibit clear morphological differences as compared with the control group. The CaOx particles observed by TEM have rhomboidal forms that are in close contact with drop-like particles of 50 to 250 nm in size that show filament-like projections suggesting the liquid nature of them (Figure 8a,b). SAED patterns confirm the precipitation of ACO structures at the post-nucleation stage. The FESEM images show the coexistence of nanocrystals with the typical pinacoid and bipyramidal forms of COT and COD hydrate phases (Figure 8c–f and Appendix A). Such crystals appear in conjunction with crystals showing pinacoid or hexagonal prism forms that aggregate adopting a tortuous shapeless structure (Figure 8c). The atypical morphologies were formed possibly due to the rapid drying of liquid precursors as a result of quenching and the drying procedure, favoring the formation of crystal shapes with liquid-like appearance.

Aggregates of nanoparticles 50 to 100 nm in size form a pseudo-dipyramidal structure, which is systematically observed when using PAA as an additive and persists up to 6000 s in the post-nucleation stage (Figure 8d, Appendix A). Such a pseudo-dipyramidal structure has a morphology similar to the final morphology of COD (Figure 8d,e). Some linear ACO aggregates remain on the surface of well-formed COD crystals (Figure 8f). Altogether, these features suggest that the formation of the pseudo-dipyramidal structures, and by extension CaOx crystals, takes place via the assembling and fusion of ACO nanoparticles along well-defined directions on the surface of the crystalline substrate, thus following a NCC route. In addition, during the aggregation of ACO, PAA induces an early preferential phase-selection towards COD. Thereby PAA favors the aggregation of ACO, likely with a proto-structure of COD, that can allow such an oriented attachment in a mesocrystal structure.

### 3.6. Influence of PAA on CaOx Precursors

Recent evidence indicates that the formation of CaOx crystals follows a NCC pathway involving the formation of ionic associates known as PNC and the formation of possibly liquid-like metastable amorphous phases, which will subsequently evolve to ACO nanoparticles and then to crystalline hydrate forms [14]. Three metastable amorphous phases have been also identified through the use of citric acid as an additive in automatic titration essay [15]. We found here that the formation of solid amorphous phases is preceded by a liquid-liquid phase separation in the absence and presence of PAA additive. The PNCs and PILP phases in solution can be stabilized by using PAA in a concentration-dependent fashion as we illustrate in Scheme 1. PAA avoids the agglomeration of CaOx precursors, by modifying the surface charge of the nanoparticles, favoring their electrostatic repulsion, which was corroborated by the Zeta potential measurements. The adsorption of PAA on the surface of ACO nanoparticles can lead to a volume restriction effect, reducing the entropy of the system, thereby limiting the interaction between nanoparticles. Indeed, the steric hindrance effect of neighboring PAA nonpolar chains, probably also reduced clustering of these nanoparticles. Then PAA leads to a delay in the nucleation of crystalline phases through electrostatic and steric stabilization processes, a process that is dependent of the utilized PAA concentration. This suggests, as well as in case of calcium carbonate, that the stability of ACO nanoparticles is greater when the concentration of the polymer is enough to cover the entire nanoparticle [35]. Our results show that CaOx crystallization follows a non-classical multistage route, which involves both the formation of PNC in unsaturated solutions and the participation of large aggregates of PNC and dense liquid precursors (PILPs) detected in supersaturated solutions. Then, PAA influences the crystallization dynamics favoring and stabilizing the formation of PILPs in the pre-nucleation stage, which leads to a lower aggregation of these entities, implying a delay in the initiation of nucleation, as observed in Figure 1. Moreover, the concentration of PAA also influences the formation of CaOx precursors that adopt a floccule or micellar conformation. High concentrations of PAA (50 and 100 mg/L) seem to cover all nanoparticles, which produces several points of union between two or more particles, leading to a bridging flocculation process. Precipitates then can reach sizes from 400 nm to 1 µm as was shown in Figure 3 and Appendix A. Furthermore, when 100 mg/L PAA was used, we found that during the first 600 s significant complexation of Ca^2+^ ions by PAA occurred, reflected by the concave shape of the initial section of the kinetic crystallization curve in the pre-nucleation stage as shown in Figure 1. The complexation of Ca^2+^ ions induced by the presence of carboxyl groups of PAA, led to a decrease in the initial saturation of the solution, which favors the delay in the initiation of the nucleation stage. However, this effect seems to be marginal, because in runs with lower PAA concentrations a significant delay in the onset of nucleation was observed, but no significant complexation was detected. In these cases, PAA inhibits nucleation by stabilizing and thus extending the half-life of CaOx precursors. Our findings indicate that the presence of PAA, independent of its concentration, favors more hydrated states of ACO because this anionic polymer sequester the excess of Ca^2+^ ions and water in the early stage of nucleation [35]. This would favor the formation and stability of PILP and will determine the amount of occluded water molecules into ACO nanoparticles when they are formed, which ultimately influences the morphology and structure of the final form of CaOx as shown in Figure 6 and Figure 7. In this sense, the presence of PAA during the formation of CaOx crystals on a nanoscopic or microscopic scale favors the formation of COD crystalline hydrate, a hydrate that has a less active role in the pathology of kidney stones, possibly due to its greater solubility. Thus, alterations in the synthesis of acidic macromolecules at the renal tubule level could play a key role in the formation of kidney stones. On the other hand, the incorporation of synthetic macromolecules with characteristics similar to PAA in the urinary tract could have a beneficial role for the prevention of this pathology. The current study provides additional evidence regarding the active role of acidic macromolecules on the NCC route of CaOx formation, the main organic component of kidney stones, in the prevention of urolithiasis. Indeed, our results suggest that calcium oxalate nucleation may occur through the densification of liquid precursor entities and subsequent aggregation, which leads to the formation of amorphous nanoparticles, allowing subsequent crystal growth. This event favors the initial formation of crystals with a nanogranular texture, as it was observed by FESEM analysis. Therefore, our results differ from the nucleation proposed by the classical theory, where the formation of the critical nucleus in homogeneous supersaturated solutions is a consequence of random collisions of its constituent ions and its growth depends on the addition of individual ions. We believe that the presence of liquid and solid precursor phases not only helps to understand the pathophysiological mechanisms of kidney stone formation but could also contribute to the understanding of the mechanisms involved in the entry of CaOx at the intracellular level into renal tubular cells, e.g., via PILP, mechanisms that have not been elucidated until now.

## 4. Conclusions

PAA as an additive on CaOx NCC pathway stabilizes PNC, liquid and solid amorphous precursors, avoiding the agglomeration of nanoparticles, which delays the start of nucleation. This study also suggests that the formation of ACO is preceded by a liquid-liquid phase separation, with the consequent partitioning of PILP precursors, which can reach nanoparticle sizes in the range of 20 to 400 nm. Moreover, PAA increases the water content of ACO nanoparticles which determines the amorphous phase character during early nucleation stages and the type of crystalline hydrate form. We found that PAA stabilizes more soluble amorphous and crystalline phases in the post-nucleation stage. The stabilization of ACO formed in the presence of PAA at the early stages of nucleation is caused by an electrostatic stabilization effect that modifies the electrical potential of the nanoparticle surface, thus preventing their agglomeration. The adsorption of PAA on the ACO nanoparticle surfaces also allows the steric stabilization of ACO through a volume restriction and steric hindrance effect, which limits the interaction among nanoparticles. We found that the adsorption effect is dependent on the concentration of PAA, which determines the amount of polymer that covers the surface of each nanoparticle. Finally, our results allow to evaluate the fundamental role that anionic macromolecules play in the pre- and post-nucleation stages. We believe that the study of the interfacial molecular recognition between acidic additive and inorganic nucleus of calcium minerals, e.g., CaOx, may help to improve the understanding of pathological mineralization that results in mammalian nephrolithiasis and supports the development of strategies to prevent or reduce the recurrence of this disease based on the incorporation of acidic synthetic macromolecules in the urinary tract.

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
