# Peer review of "Stabilization of Calcium Oxalate Precursors during the Pre- and Post-Nucleation Stages with Poly(acrylic acid)"

_nanomaterials, 2021, doi:10.3390/nano11010235_

Round 1
Reviewer 1 Report
Since the firs report on non-classical nucleation mechanism about 10 years ago, investigations on non-classical nucleation has been quite intense. This report is a very systematic investigation based on various techniques such as automatic calcium
19 potentiometric titration, TEM, SEM, XRD, FTIR. All the data obtained from macroscle to nanoscale analysis have been interpreted and understood systematically, even though atomic scale characterization is misssing. Anyway, I am happy to accept this report in the present form.
Author Response
"Please see the attachment"

Reviewer 2 Report
see alternatively enclosed file
The authors treat an interesting problem. Consequently, any advancement in the understanding of it I consider as being of huge interest both from a theoretical point of view as well as for design of medical treatments of the mentioned health problem.
The results of the paper as described in the abstract I understood as follows:
- The formation of pre-nucleation clusters of CaOx is supposed to proceed via a non-classical crystallization (NCC) pathway.
- PAA stabilizes CaOx precursors with size in the range of 20-400 nm, thereby delaying the nucleation of ACO nanoparticles.
- The results are supposed to be eventually of interest for the understanding of urolithiasis in mammals
I agree that CaOx may (or may not, this is not essential for the study performed here) nucleate via a NCC-pathway. The authors demonstrate that PAA – after nucleation did proceed - may stabilize aggregates of the specified range of 20-400 nm. However, the size of the stabilized aggregates I consider as too large to be related to critical cluster sizes determining nucleation of CaOx. So, this stabilization may prevent further growth of the already nucleated aggregates but has nothing to do with nucleation of them. The statement that PAA delays nucleation is therefore not founded by any of the results reported in this paper. In contrast, in accordance with their Scheme 1, it favors the formation of liquid-like precursor structures via colloidal coagulation. Inside these liquid-like segregated phase, new nucleation processes may occur. But these processes proceeding at the advanced stages have nothing to do with the problems the authors analyze in their paper. At this advanced stage, nucleation seems to be ro may be realized via the classical scenario. The authors should clearly distinguish which nucleation process they consider. So, the statement of the authors “we show how acidic macromolecules influence the dynamics and kinetics of CaOx crystallization by exerting control on the pre- and post-nucleation stages by delaying the crystallization of solid phases and allowing the stabilization of PNCs” I consider as only partly correct. And the main problem which has to be obviously resolved is why (as noted by the authors) “only COM and COD are found in urine, whereas COM, the most thermodynamically stable form, is the main inorganic crystal found in 42 kidney stone”. As it seems to me this problem is even not touched.
Some minor comments:
- Dillman and Meier Ref.5 analyze condensation of gases. Why this paper is cited and not papers dealing with segregation and crystallization in solutions?
- The authors state: “Biogenic mineral crystallization processes always involve the stages of nucleation and crystal growth [8]”. Question: Do the authors assume that this is a specific feature of biogenic materials?
- What is the meaning of the statement “Crystallization pathway that involves a liquid-liquid phase separation and the formation of a dense liquid phase in supersaturated solutions.”
Summarizing, the topic of analysis I consider as highly interesting. The authors have to formulate however more precisely what they have done and in which respect it can be of interest for the analysis of the formation of kidney stones.
I recommend major revision.
Author Response
"Please see the attachment"

Reviewer 3 Report
"Stabilization of calcium oxalate precursors during the pre- and post-nucleation stages with poly(acrylic acid)" presents observations of calcium oxalate nucleation in the presence of an additive, poly(acrylic acid). The work is nicely done, and is of interest primarily because the relavance of calcium oxalate in biology (eg., kidney stones) and because this system has been proposed to proceed via a non-classical nucleation route. This is an interesting manuscript that would be of interest to the readership of Nanomaterials (although may be more of more interest to the readership of Crystals). I have no major issues with the manuscript, but do have some comments prior to publication.
Mainly, I was skeptical of some of the characterization techniques for the pre-nucleation clusters that required drying, such as for the TEM analysis. The drying of the clusters likely changed the properties of the said clusters, so why do this analysis to begin with? What is learned?
Also, can the authors comment on the general criticism of some non-classical nucleation pathways, where it may be more accurate to describe the process observed by the authors as being phase separation to amorphous clusters followed by a classical nucleation process?
Author Response
"Please see the attachment"

Reviewer 4 Report
Dear Authors,
I read with great interest your paper about the effect of PAA on the stabilization of calcium oxalate polymorphs. You can find my comments directly on the copy of the paper uploaded. My only additional comment is totally personal: I'm not too fond of the intensive usage of acronyms in a text because, in my opinion, it reduces readability and fluency. For example, CC and NCC are not used so frequently in the text to justify the shortening. But this is only personal taste.
Have a nice holiday season and stay safe

Author Response
"Please see the attachment"

Round 2
Reviewer 2 Report
The authors have answered most (not all) of my questions and modified their manuscript accordingly. As I noted already, the topic of analysis I consider as highly interesting and the present paper can stimulate further research in the solution of this for sure really complex problem. I recommend publication in the present form.